# COVID-19 in the context of pregnancy, infancy and parenting (CoCoPIP) study: protocol for a longitudinal study of parental mental health, social interactions, physical growth and cognitive development of infants during the pandemic

Ezra Aydin [1,2] Staci M Weiss,[1] Kevin A Glasgow,[3] Jane Barlow,[4] Topun Austin,[5,6] Mark H Johnson,[1] Sarah Lloyd-Fox[1]

EA and SMW are joint first authors.

For numbered affiliations see end of article.

**Correspondence to**
Dr Ezra Aydin;
ea420@cam.ac.uk

## ABSTRACT

**Introduction** While the secondary impact of the COVID-19 pandemic on the psychological well-being of pregnant women and parents has become apparent over the past year, the impact of these changes on early social interactions, physical growth and cognitive development of their infants is unknown, as is the way in which a range of COVID-19-related changes have mediated this impact. This study (CoCoPIP) will investigate: (1) how parent's experiences of the social, medical and financial changes during the pandemic have impacted prenatal and postnatal parental mental health and parent–infant social interaction; and (2) the extent to which these COVID-19-related changes in parental prenatal and postnatal mental health and social interaction are associated with fetal and infant development.

**Methods and analysis** The CoCoPIP study is a national online survey initiated in July 2020. This ongoing study (n=1700 families currently enrolled as of 6 May 2021) involves both quantitative and qualitative data being collected across pregnancy and infancy. It is designed to identify the longitudinal impact of the pandemic from pregnancy to 2 years of age as assessed using a range of parent- and self-report measures, with the aim of identifying if stress-associated moderators (ie, loss of income, COVID-19 illness, access to ante/postnatal support) appear to impact parental mental health, and in turn, infant development. In addition, we aim to document individual differences in social and cognitive development in toddlers who were born during restrictions intended to mitigate COVID-19 spread (eg, social distancing, national lockdowns).

**Ethics and dissemination** Ethical approval was given by the University of Cambridge, Psychology Research Ethics Committee (PRE.2020.077). Findings will be made available via community engagement, public forums (eg, social media,) and to national (eg, NHS England) and local (Cambridge Universities Hospitals NHS Foundation Trust) healthcare partners. Results will be submitted for publication in peer-reviews journals.

## STRENGTHS AND LIMITATIONS OF THIS STUDY

⇒ This is a new cohort of families being followed from prenatal to postnatal (up to 18 months) during the COVID-19 pandemic.
⇒ The study involves the collection of quantifiable parent-report data to identify the short-term and long-term influences of the pandemic on key aspects of infant development.
⇒ The study also has a range of open-ended questions for qualitative analysis aimed at exploring familial experiences in more detail.
⇒ The data are being collected online and is therefore limited to self-report and parent-report measures, with no direct assessment of child development and parental mental health.
⇒ Although the sample of families being recruited are diverse in their indices of multiple deprivation and geographical location, they may not be fully representative of the wider population.

## INTRODUCTION

The COVID-19 pandemic has resulted in an unprecedented challenge to existing medical, social and economic institutions, raising the risk for exposure to adversity for families expecting or parenting babies akin to prior natural disasters, war, or other periods of hardship.[1] Infants born during periods of social disruption and disease are noted for more restricted intrauterine growth, smaller birth size, and higher lifetime incidence of chronic medical conditions such as type-II diabetes, suggesting a role for fetal programming of endocrine dysfunction and metabolic regulation.[2][3] Consequently, parents who were expecting or delivered babies amidst

pandemics may experience enduring impacts on their well-being, compounded by parenting children at elevated risk for stress-related changes in the early development and lifelong health of children.[4 5] Conceptual frameworks have been advanced regarding the lifelong effects of adversity in pregnancy and early childhood. As specified in the developmental origins of health hypothesis, parental stress interacts with environmental exposures (eg, nutrition, pollution), to influence the maternal-fetal physiological feedback (as indicated by hormonal and inflammatory biomarkers).[6 7] Parental behaviour and availability in the early postnatal period (eg, parenting interaction and sensitivity) in turn shapes later biological, physical and neurodevelopmental outcomes.[8] In wake of the COVID-19 pandemic, toxic stress-informed frameworks for promotion of parental mental health during pregnancy have been expanded to include postnatal mental health and healthy parent–infant attachment amidst disrupted access to direct caregiving support.[9] Further, the ecobiodevelopmental framework illustrates how modifiable early environmental influences—such as unemployment, family poverty and access to healthcare—can impart an enduring effect on children's stress physiology and genetic expression.[6 10 11] An associated framework put forward by Nelson and Gabard-Durnam[12 13] suggests that we should view adversity as a violation of the expectable environment, with emphasis placed on the magnitude of this impact being greater during critical periods of brain development (such as the first 1000 days from conception to toddlerhood).

Emerging work is documenting the long-term implications of adversity related to the current pandemic including, for example, biological (ie, COVID-19 infection), acute environmental (ie, temporary unemployment and psychosocial influences (ie, impoverished, or atypical social environment)).[14–16] The social distancing restrictions and national lockdowns that were put in place to mitigate COVID-19 transmission have had a range of secondary consequences impacting the psychological well-being of pregnant women and new parents and the postnatal psychosocial environment that the infant is born into.[17–19] The shifts in socialisation, stress and socioeconomic position associated with COVID-19 public health guidance may have exacerbated the feelings of vulnerability, health vigilance and isolation associated with the adjustment to parenting. Heightened anxiety and depression were reported during the national lockdown in the UK,[20] with expectant and new mothers and fathers experiencing unique physical and psychological stressors[21 22] as well as constrained access to resources, especially with regard to family and caregiving support.[23] However, little is currently known about the impact of these COVID-19-related changes on the development of the infant. The current study aims to address this evidence gap by exploring the relationship regarding the family's reported experiences of these changes in terms of their impact on their prenatal and postnatal mental health, and interaction with their infant, and the potential subsequent impact of these changes on the infant's physical, sensory, affective, and cognitive development.

## The secondary impact of COVID-19 on pregnant women

The COVID-19 pandemic has been the biggest public health emergency for over a century, necessitating extreme measures at a societal level to mitigate against death and prevent acute health services from being overwhelmed. These changes have led to a number of secondary consequences (ie, increased caregiving demands for children and family members; isolation from family and community due to social distancing; job loss; financial hardship and increased interpersonal stressors or relationship violence), having a disproportionate and significant impact on women of childbearing age.[23 24] In the UK, the impact of the pandemic on this group of women has also been exacerbated by National Health Service (NHS) guidance that was produced in response to the national lockdown restrictions,[25] in which hospital-based midwifery services placed limitations on partners being present during ultrasound visits and birth. In addition, most community-based services were discontinued, other than antenatal contact and new baby visits, all of which were required to be provided virtually unless otherwise indicated[26] with all other contacts being assessed and stratified according to vulnerability or clinical need (eg, maternal mental health).

These changes not only affected the capacity of practitioners to support women during the perinatal period at a time of significantly heightened stress/distress[4] but also resulted in significant regional variations in access to healthcare and advice for expectant mothers across the UK. The changes have been associated with (1) a four-fold increase in stillbirths attributed to lack of preventive antenatal care, (2) birthing partners denied access to the hospital for the birth or asked to leave immediately following the birth, and (3) limited access to babies admitted to neonatal intensive care.[27 28] The NHS has also reported a reluctance on the part of parents to attend postnatal GP checks, due to parental attitudes related to COVID-19 infection (Institute of Health Visiting).

## The impact of COVID-19 on parental mental health during pregnancy

Several online surveys conducted during the first national lockdown indicated that there was a significant increase in antenatal anxiety both in terms of pandemic-related pregnancy stress associated with feeling unprepared for birth due to the pandemic, and stress related to fears of perinatal COVID-19 infection, with one large US survey (n=4451) showing that around 30% of pregnant women experienced both types of stress.[29] Another US survey (n=2740) that examined wider sources of stress showed that more than half of women reported increased stress in relation to concerns about food running out (59.2%, n=1622), losing a job or household income (63.7%, n=1745), or loss of childcare (56.3%, n=1543). More than a third reported increased stress about conflict

between household members (37.5%, n=1028); and 93% (n=2556) reported increased stress about getting infected with COVID-19.[30]

A number of online cross-sectional surveys also found significantly increased rates of anxiety and depression, based on the use of self-report standardised measures (eg, Edinburgh Postnatal Depression Scale; Hospital Anxiety and Depression Scale). For example, a cross-sectional survey of 1987 pregnant women in Canada in April 2020, found substantially elevated anxiety and depression symptoms, compared with similar prepandemic pregnancy cohorts; 37% reported clinically relevant symptoms of depression, and 57% reported clinically relevant symptoms of anxiety.[31] A second Canadian study found that a cohort of pregnant women, who were recruited during the COVID-19 pandemic, were twice as likely to present clinically significant levels of depressive and anxiety symptoms compared with a cohort of pregnant women recruited prior to the pandemic.[32] Early evidence in the UK similarly suggests that the impact on the mental health of pregnant women has been significant with heightened anxiety and depression being reported during the national lockdown (levels of mental distress rising from 18.9% (2018–2019) to 27.3% in April 2020, 1 month into the national lockdown).[33]

This is of concern because there is consistent evidence to suggest that anxiety and depression in pregnancy can have a long-term impact on child development. For example, traumatic birth experiences amidst the changing public health situation or COVID-19 infection in the household have been associated with unusual parent–infant bonding.[34] Recent systematic reviews found that antenatal anxiety is associated with a range of adverse perinatal outcomes, including, for example, premature delivery and low birth weight,[35] in addition to a range of negative child outcomes that can persist into late adolescence, including an increased risk of child behaviour problems.[36]

### The impact of COVID-19 on parental mental health, parent–infant interaction, and infant early environment

There is growing evidence about how the pandemic and lockdown-related stressors impacted parental mental health during the postnatal period. Studies of postnatal depression suggest a similar picture to that prenatal, with around half of mothers caring for babies born during 2020 in the UK, reporting feeling down, lonely and worried, with mental health symptoms exacerbated in mothers who travel to work, had a baby born prematurely or were from a lower income household.[37 38] One Australian study that examined all online perinatal support forum posts related to COVID-19, from women between 27 January to 12 May 2020, showed that the content was predominantly negative, with around 63% being very or moderately negative. Negative words that were frequently used in the 831 posts included: 'worried' (n=165, 19.9%), 'risk' (n=143, 17.2%), 'anxiety' (n=98, 11.8%), 'concerns' (n=74, 8.8%), and 'stress' (n=69, 8.3%).[38] Similarly,

first-time fathers who became parents during the onset of the pandemic in Italy reported greater stress than those with older children, and a study of fathers in Israel found that those who reported greater pandemic and parenting stress were more likely to report dysfunctional interactions with their infant and identify their baby's temperament as difficult.[22 39]

Anxiety and depression in the postnatal period have been shown to affect the development of the infant because of the impact on the parent's interactions with their baby. For example, depressed mothers have been shown to be less sensitively attuned to infants, and less affirming and more negating of their experiences with their infant.[40] Babies of depressed mothers can exhibit deficits in their interpersonal functioning, such as less affective sharing, lower rates of interactive behaviour, poorer concentration, increased negative responses with strangers, and reduced secure attachment at 12 and 18 months.[41 42] Children of a mother who had postnatal depression are 42% more likely to experience depression by age 16.[43] The prenatal and postnatal mental health of caregiving partners (including fathers) also appears to influence caregiver-infant interaction.[44 45]

Although anecdotal reports of lockdown and work from home suggest more physical, social and material support for primary caregivers (from co-parents, fathers, non-marital partners, grandparents), and parents being more present and involved in caregiving for babies born during the pandemic compared with older siblings, the evidence to support this is currently lacking. Furthermore, it seems likely that any such benefits are socially determined: socioeconomically deprived families have unique vulnerability to the isolation of parenting amidst the pandemic.[46 47] Lower income families reported more frequent issues with breast feeding, higher incidence of postnatal depression, and difficulty accessing caregiving and social support that might ameliorate the demands of caring for newborns.[48–50]

In addition, limited exposure to infant peers or diverse social partners other than household members, or exposure to members of the public wearing masks, might confer different strategies for infant looking and communicative bids during social interactions. While ongoing research is examining how the pandemic and its cascading effect on early contextual factors are affecting children ages 8–36 months, there is a gap within the research observing earlier instances of development during the period referred to as the 'baby blind-spot' (from pregnancy to 2 years old age).[18 47] This study aims to bridge this gap by integrating pandemic-specific changes in parental mental health and COVID-19 induced social guidance as a unique context for infant development.[51]

There also remains more questions than empirical answers at the present time regarding how 'stay at home' orders, lack of access to social support from family members and pandemic-specific stressors might have affected women at risk for domestic violence which was found to have increased significantly during the

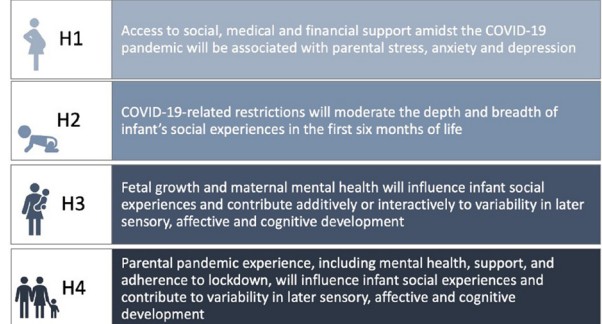

**Figure 1** COVID-19 in the Context of Pregnancy, Infancy and Parenting (CoCoPIP) study four key hypotheses.

pandemic,[21 52] and women heavily impacted by unemployment due to their over-representation in the retail, caregiving and hospitality workforce.[52]

It is essential for longitudinal studies to address the impact of COVID-19 guidance and restrictions on the long-term development of the child,[53] particularly in the UK as an example of a nation with an above-average COVID-19 mortality rate, high-income disparity and centralised healthcare system with a fairly uniform government response across regions with varying rates of infection. As such, our sample could serve as a test of the developmental programming hypothesis by assessing the extent to which a range of key domains of child development have been influenced by changes in stress and infant social exposure from pregnancy to early infancy, arising from the social restrictions in place at different points during the pandemic. This study has two main goals: (1) to examine how parent's experiences of the social, medical and financial changes during the pandemic have impacted prenatal and postnatal parental mental health and parent–infant social interaction; and (2) to investigate the extent to which these COVID-19 related changes in parental prenatal and postnatal mental health and social interaction are associated with fetal and infant development.

## STUDY DESIGN

The COVID-19 in the Context of Pregnancy, Infancy and Parenting (CoCoPIP) study is a national online survey being carried out in the UK, which was widely advertised from July 2020, that continues to actively recruit families for participation. The research comprises a mixed-method study collecting data inclusive of both: (1) validated physical and psychological assessments, and (2) open-ended questions to allow the participant to elaborate on their experience in their own words. The large sample collected enables us to use data-driven (lasso, Bayesian with infants born prepandemic as dictating priors) and hypothesis-driven approaches to assess if COVID-19 provides a model for how individual factors (maternal mental health, birth timing, caregiving) interact with institutional factors (government lockdowns, social and financial support, etc).

The CoCoPIP Study addresses four key hypotheses (H1-4) (see figure 1). Our variable selection and sequential building of hypotheses embed our key frameworks: (1) examining parental mental health in light of stress and social, financial and contextual factors (ecobiodevelopmental model); (2) how infant sensory processing pertains to caregiving and social exposure of infants, relative to lockdown/COVID-19 transmission during infant's birth, family COVID-19 vigilance and parenting anxiety (early expectable environment); (3) the interaction of maternal mental health and fetal growth measures as longitudinal predictors of infant cognitive outcomes (developmental origins of disease hypothesis); (4) finally, encompassing the social, financial and contextual factors which impact parental mental health to shape infant temperament and sensory processing, accounting for early infant caregiving and social environment (developmental programming). Ultimately, our research programme can demonstrate support (or lack thereof) for extending developmental programming-based frameworks beyond child physical health (insulin resistance, stature, etc)[54 55] and cognitive outcomes,[56–58] to explain variability in proximal domains such as infant affective, social and sensory capacities.

## Eligibility criteria and recruitment strategies

Eligibility criteria for the study is expectant parents (at any stage of pregnancy) or parents of an infant between the ages of 0–6 months. Either parent can take part, with questionnaires being adapted to the parents' status as mother or father. The study is open to parents who had a baby within 6 months prior to the first period of lockdown in the UK (23 March 2020) as well as continuing to collect information from parents during the current and future changes in COVID-19-related health and societal restrictions.

For optimal national representation across the UK, recruitment strategies include (1) targeting NHS antenatal classes and National Childbirth Trust (NCT) groups identified nationally, with an emphasis on areas of low socioeconomic status using the government indices of multiple deprivation (IMD) and rural areas without access to NCT groups, (2) partnering with NHS/National Institute for Health Research collaborative sites and charity and policy group partners (eg, The Brazelton Centre UK, Centre for Health and the Public Interest) to widen knowledge of the survey, (3) posting online via social media platforms (eg, Twitter) and public sharing to facilitate snowball sampling and (4) targeting populations experiencing increased local lockdown measures as, and when, COVID-19 rates and related policy change across the UK. While recruitment efforts have been focused on the UK, the survey is currently open to all expectant and new families worldwide. Participation in the survey is incentivised using the offer of a chance to win a £100 digital gift card (on receipt participants are able to select from either an Amazon® or one4all® gift card). A prize is drawn for every 100 participants who complete the survey,

giving a 1/100 chance of winning at each time point that they complete.

## Patient and public involvement

The study was designed with input from the public, particularly pregnant and new parents who had an infant during the onset of the pandemic from March to May 2020. Input included aspects such as wording of questions and ease of completing questionnaires (both visually and in length). Results will be disseminated to study participants through various social media platforms that participants are given links to during initial recruitment and in subsequent follow-up correspondence.

## Power calculation

To ensure sufficient power for the study, statistical power calculations (G*Power adapted for regression) based on three outcomes, up to three predictors and four covariables, estimated a minimum sample size for Hypothesis 1 of n=400 (small effect, $f^2$=0.02). Statistical power calculations were based on a study of sociodemographic control variables, traumatic event impact scale and pregnancy-specific anxiety.[37] In the same manner, for hypothesis 2, a minimum sample size of n=800 (small effect, n=400 infants × 2 postnatal timepoints) is required, based on ongoing analyses by our group on parent–infant social interaction data.[38] For hypothesis 3 and 4, we will minimise data loss using post-hoc assignment of families to an accelerated longitudinal design—this requires a minimum sample size of n=500 (small effect) using a study on acute disasters, parental mental health and infant development.[39] With timing and cross lag accounted for, and attrition rate of 30%—assuming current pattern of 80% of parents' consenting to be contacted again (as indicated by our pilot survey)—a minimum cohort sample of n=1500 is required.

## Study measures

The online survey is logic-dependent and adaptive, only showing questions relevant to the parent's current situation (eg, antenatal or with an infant of 2 months of age) in relation to the following six time points: the second and third trimester of pregnancy; infant aged 0–3 months, infant aged 3–6 months; and toddler aged 12- and 18 months. The following data are collected: (1) parental mental health and attitudes, (2) healthcare access and support during pregnancy and birth, (3) fetal physical development and infant social and cognitive development, (4) direct impact of COVID-19 on daily lives and lastly (5) developmental outcomes in infants born during the pandemic. Table 1 provides an overview of the measures used timepoints (for a detailed summary of the measures and questionnaires used within the survey see online supplemental file 1, online supplemental table 1).

## Follow-ups and reminders

Participants are invited to take part in a follow-up survey at the end of the initial survey. Where they consent to this, they are contacted via email containing a link to the separate online survey. The follow-up survey has been condensed to include follow-up questionnaires only (*see table 1* and figure 2 for participant follow-up flow chart). The appropriate time for follow-up is calculated based on the ages (infant or fetal gestation) provided by the participant at initial recruitment (see figure 2 for project timeline).

## DATA ANALYSIS PLAN

The Statistical Analysis Plan was developed based on the UK Dept of Health/Medical Research Council Clinical Trials Toolkit and NHS epidemiological study designs, with details outlined per the standards for random control trials and clinical trials.[59 60]

## Quality control

Ongoing quality control is evaluated biweekly. All data are checked for accuracy and invalid data are removed. Study data are collected and managed using Research Electronic Data Capture (REDCap®) tools hosted at the University of Cambridge.[61] REDCap® is a secure, web-based software platform designed to support data capture for research studies, providing (1) an intuitive interface for validated data capture; (2) audit trails for tracking data manipulation and export procedures; (3) automated export procedures for seamless data downloads to common statistical packages; and (4) procedures for data integration and interoperability with external sources. Personal data (eg, caregiver DOB, email address) is stored securely within a password encrypted electronic databased isolated from the research data. Access to the data is fully audited to ensure data security is governed by a management team and in compliance with ethical guidelines.[62]

## Analysis plan

Overall, we aim to identify which stress-associated moderators (ie, loss of income, COVID-19 illness, local access to ante/postnatal support) impact significantly on parental mental health, and in turn, infant development. Further to the plans outlined for each aim below, tests of normality and sensitivity analyses (comparing observed values and imputed missing values) will be conducted. Non-linear tests of significance and interpolation approaches will be applied where appropriate.

To address hypothesis 1 (see figure 1), a combination of quantitative and qualitative analyses will be undertaken. Structural equation modelling (SEM) and hypothesis-driven regressions will explore how multiple aspects of prenatal and postnatal family support (social, financial and health) are associated with latent outcomes of stress (parenting anxiety and pandemic-related stress) and latent outcomes of mental health symptoms. An inductive approach to data analysis will be undertaken[63] for the open-ended qualitative data, and we will use NVivo (QSR International Pty) to code the data. Following this, several approaches including thematic,[63] sentiment and context content analysis will be undertaken using a natural

**Table 1** A summary of assessments and questionnaires used separated by time point

| | Timepoint | | | | | |
| --- | --- | --- | --- | --- | --- | --- |
| | Pregnancy | | Infant | | Toddler | |
| | 1 | 2 | 3 | 4 | 5 | 6 |
| **Section A: consent and participant background information** | | | | | | |
| Consent* | x | x | x | x | | |
| Demographics* | x | x | x | x | | |
| Income and employment status | x | x | x | x | | |
| **Section B: pregnancy measures** | | | | | | |
| Fetal growth measures and pregnancy | x† | x‡ | | | | |
| Healthcare support and access | x | x | x | x | | |
| Antenatal emotional attachment scale (AEAS) | x | x | | | | |
| Pregnancy related anxiety questionnaire revised (PRAQ-R) | x | x | | | | |
| **Section C: infant birth and development measures** | | | | | | |
| Birth information§ | | | x | x | | |
| Infant behaviour questionnaire (IBQ) | | | | x | x | |
| Infant toddler sensory profile (ITSP) | | | x | x | x | |
| Infant-related anxiety | | | x | x | | |
| Face-to-face interaction index | | | x | x | | |
| **Section D: toddler development measures** | | | | | | |
| Ages and stages | | | | | x | |
| Oxford CDI | | | | | x | |
| Q-CHAT | | | | | | x |
| Vineland parent and caregiver form | | | | | | x |
| **Section E: parental mental health and support measures** | | | | | | |
| State Trait Anxiety Index – State (STAI-S) | x | x | x | x | x | x |
| Caregiving, social interaction and support questionnaire | | | x | x | | |
| Stressful life events questionnaire | | | | | x | x |
| **Section F: parenting and family measures** | | | | | | |
| Parenting reflective functioning questionnaire (PRFQ) | | | | | x | |
| Comprehensive early childhood parenting questionnaire (CECPAQ) | | | | | x | |
| **Section G: COVID-19 Impact** | | | | | | |
| COVID-19 situational influences | x | x | x | x | | |
| COVID-19 health report | x | x | x | x | | |
| COVID-19 concern and event impact scale | x | x | x | x | | |
| Social distance impacts | x | x | x | x | | |
| Vaccine | x | x | x | x | | |
| Difficulties in Emotion Regulation Scale (DERS) | | | | | x | |

*Participants will only be asked to complete this section once, when they initially join the study. The study can be joined at any timepoint. Those eligible will be asked if they wish to participate longitudinally.
† Only physical questions in relation to second trimester scan.
‡Only physical questions in relation to third trimester scan.
§Will only be asked to complete this section once.

language processing approach[64] (machine learning) to identify forms of social, medical and financial support in relation to the valence of parental attitudes. Regression analyses will then be conducted to understand the directional relationship between resulting latent factors (qualitative responses and quantitative data) and parental mental health.

To address hypothesis 2, Bayesian non-linear regressions will be used to explore how COVID-19 related restrictions during an infant's birth altered infant social

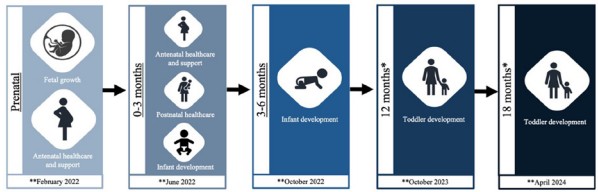

**Figure 2** Flow diagram of COVID-19 in the Context of Pregnancy, Infancy and Parenting (CoCoPIP) study follow-up participation. A participant can join the study at any of the above three underlined timepoints. *Participants are only followed up at these two timepoints if they have participated in at least two previous timepoints. **Projected participant follow-up completion dates.

exposure with caregivers and non-household social partners, and the impact of these in turn on infant processing of sounds, sights and social stimuli. The dependent variables will be derived from scores of the infant toddler sensory profile, a standard assessment of infant self-regulation and responsiveness to their environment. On the first level, the timing of an infant's birth will be coded based on whether a lockdown or no lockdown was imposed by the government, as well as coding for more specific shifts in the UK government public health policy and 'unlocking' guidance from July 14 2020 to July 19 2021. Additional COVID-19 factors may be entered, including suspected or positive cases in loved ones, parental COVID-19 concern, pandemic-related parenting anxiety and parent-reported adherence to lockdown. On the next level, infant's social exposure will draw from caregiver-reported of their frequency of face-to-face interactions with their baby, as well as their baby's exposure to

social partners from outside the household, in-person, at a distance and online. On the third level, family sociodemographic factors, such as the number of family members in each household, family income, ethnicity and high-risk health conditions will be included.

To address hypothesis 3, linear regressions will be used to explore the influence of maternal mental health longitudinally on the developing offspring across prenatal to postnatal life: from fetal (12 weeks/20 weeks gestational age) to 18 months of age. After testing for normality and transforming variables accordingly, standardised z-scores will be created from the collected fetal growth measurements (ie, head circumference, femur length and abdominal circumference), which will then be transformed into a composite score accounting for gestational age and/or estimated fetal weight at time of scan to be used for analysis. Z-scores will also be computed and used where appropriate within the analysis (eg, infant-toddler sensory profile).

To address hypothesis 4, outputs from hypothesis 1 and 2 (impact of COVID-19 on parental mental health and infant social interactions) will be nominated using lasso regression coefficients in relation to longitudinal social and cognitive child development domains (assessing language, motor sensory and the early emergence of developmental conditions) of the infant/toddler across the 0–18 months of life using SEM, applying full information maximum likelihood to account for missingness and to identify developmental-hypothesis driven clusters of affected families by factors such as birth timing, individual family stressors and pandemic restrictions during survey.

### Current cohort description and demographics

Initiated in July 2020, this study is ongoing with n=1700 families currently enrolled (6 May 2021). Parents can consent to complete the questionnaire up to six times during pregnancy/parenting until their infants are 18 months of age. For those participants who contribute more than one time point (between antenatal and postnatal timepoints≤6 months), an invitation is issued for a follow-up to assess their toddler's development when aged 12 and 18 months (see figure 2 for study flow chart).

To date 1700 of families have participated in at least one time-point of the study, with 641 families joining at time points 1–2, 372 families at time-point 3, and 687 families at time-point 4 (see figure 2 and table 1 for time-points). Sixty-one per cent of these families have consented to completing the subsequent follow-up sections of the study. To date 97.4% of respondents, identify as mothers, 2.3% as fathers and 0.3% as another parent or caregiver, with the majority of participating families disclosing their ethnicity as white (89.2%). Those participating families who are from the UK have their household information (ie, household income, location of participating families, index of multiple deprivation and respondent's education level) described in figure 3.

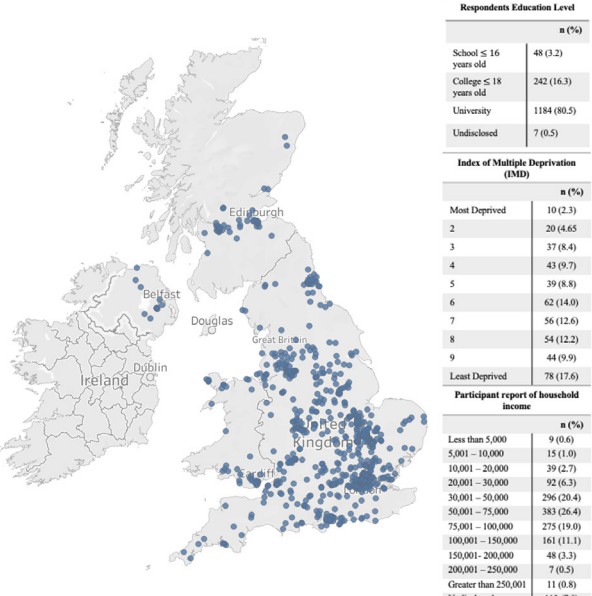

| Respondents Education Level | |
| --- | --- |
| | n (%) |
| School ≤ 16 years old | 48 (3.2) |
| College ≤ 18 years old | 242 (16.3) |
| University | 1184 (80.5) |
| Undisclosed | 7 (0.5) |

| Index of Multiple Deprivation (IMD) | |
| --- | --- |
| | n (%) |
| Most Deprived | 10 (2.3) |
| 2 | 20 (4.65) |
| 3 | 37 (8.4) |
| 4 | 43 (9.7) |
| 5 | 39 (8.8) |
| 6 | 62 (14.0) |
| 7 | 56 (12.6) |
| 8 | 54 (12.2) |
| 9 | 44 (9.9) |
| Least Deprived | 78 (17.6) |

| Participant report of household income | |
| --- | --- |
| | n (%) |
| Less than 5,000 | 9 (0.6) |
| 5,001 – 10,000 | 15 (1.0) |
| 10,001 – 20,000 | 39 (2.7) |
| 20,001 – 30,000 | 92 (6.3) |
| 30,001 – 50,000 | 296 (20.4) |
| 50,001 – 75,000 | 383 (26.4) |
| 75,001 – 100,000 | 275 (19.0) |
| 100,001 – 150,000 | 161 (11.1) |
| 150,001– 200,000 | 48 (3.3) |
| 200,001 – 250,000 | 7 (0.5) |
| Greater than 250,001 | 11 (0.8) |
| Undisclosed | 113 (7.8) |

**Figure 3** Bubble map depicting spread of participations location in the UK (if postcode was provided) with respondent's education level, Index of Multiple Deprivation (IMD) and household income breakdown reported on the right.

## ETHICS AND DISSEMINATION

Ethics approval for the survey was given by the University of Cambridge, Psychology Research Ethics Committee (PRE.2020.077). All respondents are required to be over the age of 18 years and give electronic informed consent. Caregivers agreeing to be followed up longitudinally give consent at each timepoint and are made aware that their participation can be stopped at any time within the study. Permissions have been obtained from participants to ensure anonymised data can be made available on open-source platforms.

A proactive dissemination pathway has been established from the outset. We will engage with policy stakeholders (health practitioners/Department of Health) and social media platforms to create discussion around this topic. Dissemination of findings will be via public forums (ie, social media, media, collaborators family dissemination pathways) and at the national (ie, NHS England/NHS Improvement, Royal College of Paediatrics and Child Health, Centre for Health and the Public Interest) and local (Cambridge Universities Hospitals NHS Foundation Trust) level. Data will continue to be disseminated throughout the period of the study to promote discussion and raise the profile of the population identified as being one of the most vulnerable and neglected during the pandemic.

To date, qualitative responses from the first 5 months of data collection have been analysed to explore parents experiences of being pregnant in relation to healthcare access during the pandemic.[19] This was conducted using thematic and sentiment analysis. The initial findings suggest that a range of adverse effects have been experienced by expectant parents in the UK relating to changes in antenatal support and healthcare appointments in response to governmental guidance with regard to social distancing. These findings point to an urgent need to better address the unique healthcare needs of each pregnant woman going forward.

### Data sharing plan

Questionnaires and study goals were made available on request using the Open Science Foundation platform in July 2020 and made public at https://osf.io/m7zuw/ in August 2020. Study protocol, follow-up questionnaires and statistical analysis code will be uploaded and shared to facilitate data sharing and collaboration, in accordance with Research, Innovation and Science Policy Experts EU principles.[65] Qualitative data generated and analysed during the study will not be made publicly available due to ethical and privacy restrictions; however, researchers can submit a research proposal to the Data Sharing Management Committee to request access and collaboration.

### Author affiliations
[1]Department of Psychology, University of Cambridge, Cambridge, UK
[2]Department of Psychiatry, Columbia University, New York City, NY, USA
[3]Department of Education, University of Cambridge, Cambridge, UK
[4]Department of Social Policy, University of Oxford, Oxford, UK
[5]Rosie Hospital, Cambridge, UK
[6]NIHR Cambridge Biomedical Research Centre, Cambridge, UK

**Acknowledgements** We are extremely grateful to all those families who gave their time to participate and to Esther Adememo, Maddie Walton and Zahra Khan who worked on the CoCoPIP study during their undergraduate and master's studies at the University of Cambridge.

**Contributors** EA: conceptualisation, methodology, writing - original draft. SMW: conceptualisation, methodology, writing - original draft. KAG: methodology, visualisation, writing - review & editing. TA: methodology, supervision, writing - review & editing. MHJ: supervision, funding acquisition, writing - review & editing. JB: supervision, writing - review & editing. SL-Fox: conceptualisation, methodology, supervision, funding acquisition, writing - review & editing.

**Funding** This research was funded by a Medical Research Council Programme Grant MR/T003057/1 to MJ, and an UKRI Future Leaders fellowship (MRC grant MR/S018425/1) to SLF. The views expressed are those of the authors and not necessarily those of the MRC or the UKRI. The NIHR Cambridge Biomedical Research Centre (BRC) is a partnership between Cambridge University Hospitals NHS Foundation Trust and the University of Cambridge, funded by the National Institute for Health Research (NIHR), TA is supported by the NIHR Cambridge Biomedical Research Centre (BRC). TA is also supported by the NIHR Brain Injury MedTech Co-operative.

**Disclaimer** The views expressed are those of the author(s) and not necessarily those of the NIHR or the Department of Health and Social Care.

**Map disclaimer** The depiction of boundaries on this map does not imply the expression of any opinion whatsoever on the part of BMJ (or any member of its group) concerning the legal status of any country, territory, jurisdiction or area or of its authorities. This map is provided without any warranty of any kind, either express or implied.

**Competing interests** None declared.

**Patient and public involvement** Patients and/or the public were involved in the design, or conduct, or reporting, or dissemination plans of this research. Refer to the Methods section for further details.

**Patient consent for publication** Not applicable.

**Provenance and peer review** Not commissioned; externally peer reviewed.

**ORCID iD**
Ezra Aydin http://orcid.org/0000-0003-4845-053X

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
