## [Reviewer comments · BMJ Open]

ARTICLE DETAILS

TITLE (PROVISIONAL)	The COVID in the Context of Pregnancy, Infancy and Parenting (CoCoPIP) Study: protocol for a longitudinal study of parental mental health, social interactions, physical growth, and cognitive development of infants during the pandemic.
AUTHORS	Aydin, Ezra; Weiss, Staci; Glasgow, Kevin; Barlow, Jane; Austin, Topun; Johnson, Mark H.; Lloyd-Fox, Sarah

VERSION 1 – REVIEW

REVIEWER	Ye, Jiancheng Northwestern University Feinberg School of Medicine
REVIEW RETURNED	06-Sep-2021

GENERAL COMMENTS	This study proposed a protocol to investigate parent's experiences of the social, medical, and financial changes during the pandemic, the extent to which these COVID-related changes in parental pre and postnatal mental health, and social interaction with fetal and infant development. It's a very important topic and the authors did great work on this study. My considerations are indicated below: • The authors may provide a framework for this study protocol.• Table 2 is very important because it provided all the measures to be collected. The authors could provide all the questionnaires in the supplementary materials rather than just provide some example items. Because some items are from existing questionnaires, but some are not clear.• It would be helpful to demonstrate the detailed statistical analysis plan, which is missing in the manuscript.• There are some recent papers discussing the intersection of parental and infant mental health, the authors may refer to the articles below to discuss some evidence and opinions in the DISCUSSION section: McVety, C.C., 2021. Effects of Parental Mental Health on Children in a Worldwide. Ye, J., 2020. Pediatric mental and behavioral health in the period of quarantine and social distancing with COVID-19. JMIR pediatrics and parenting, 3(2), p.e19867.
---

REVIEWER	Vismara, Laura University of Cagliari
REVIEW RETURNED	27-Sep-2021

GENERAL COMMENTS	I would like to thank the Editor and the Authors for the opportunity to review the protocol entitled: "The COVID in the Context of Pregnancy, Infancy and Parenting (CoCoPIP) Study: protocol for a longitudinal study of parental mental health, social interactions,
--

	physical growth, and cognitive development of infants during the pandemic”. I find the contribution of relevant contents for the Journal. Therefore, I recommend this protocol to be published. Nonetheless, I think that the manuscript needs some minor revisions. In the introduction, the Authors refer to parental mental health, but restrict their comments solely on maternal depression and anxiety; if they talk about parental, fathers’ studies should be included; in addition, mental health issues during Covid are far more complicated during the perinatal period (see and add: Ahmad, M., & Vismara, L. (2021). The Psychological Impact of COVID-19 Pandemic on Women’s Mental Health during Pregnancy: A Rapid Evidence Review. International Journal of Environmental Research and Public Health, 18(13), 7112. Liu, C. H., Erdei, C., & Mittal, L. (2021). Risk factors for depression, anxiety, and PTSD symptoms in perinatal women during the COVID-19 Pandemic. Psychiatry research, 295, 113552). It would be important to make explicit the subtended theoretical model, that has guided the choice of the study variables, methods and analyses and will offer the key of data interpretation. Please, specify why the Authors have chosen those specific time points. How will the control group will be constituted?
--	--

VERSION 1 – AUTHOR RESPONSE

Reviewer 1:

1. The authors may provide a framework for this study protocol.

We have expanded the theoretical framework for the study protocol, including not only the developmental origins of disease hypothesis but also elements of developmental/fetal programming and early expectable environment, as described by the ecobiodevelopmental model.

Page 1: *“Infants born during periods of social disruption and disease are noted for more restricted intrauterine growth, smaller birth size, and higher lifetime incidence of chronic medical conditions such as Type-II diabetes, suggesting a role for fetal programming of endocrine dysfunction and metabolic regulation [2,3]...”*

“Conceptual frameworks have been advanced regarding the lifelong effects of adversity in pregnancy and early childhood. As specified in the developmental origins of health hypothesis, parental stress interacts with environmental exposures (e.g., nutrition, pollution), to influence the maternal-fetal physiological feedback (as indicated by hormonal and inflammatory biomarkers) [6,7].”

“Further, the ecobiodevelopmental framework illustrates how modifiable early environmental influences - such as unemployment, family poverty and access to healthcare - can impart an enduring effect on children’s stress physiology and genetic expression [6,10,11]. An associated framework put forward by Nelson and Gabard-Durnam [12,13] suggests that we should view adversity as a violation of the expectable environment, with emphasis placed on the magnitude of this impact being greater during critical periods of brain development (such as the first 1000 days from conception to toddlerhood).”

- 2. Table 2 is very important because it provided all the measures to be collected. The authors could provide all the questionnaires in the supplementary materials rather than just provide some example items. Because some items are from existing questionnaires, but some are not clear.**

We have provided an output of all questionnaires within the supplementary materials as requested. However, please note where the questionnaire is standardised this has not been included. This has been made clear in the legend of Table 2.

- 3. It would be helpful to demonstrate the detailed statistical analysis plan, which is missing in the manuscript.**

Thank you to the reviewer for their comment, and accordingly we have elaborated further detail within our existing section described as “Data Analysis Plan (DAP)” which included an overview of analytic techniques for each hypothesis. When planning the format of this paper, we decided to take the approach recommended for epidemiological study designs (as the field is lacking a consensus of guidance for SAPs for research studies that are not trial based) and integrated our plans within the main protocol paper (see Hemming et al., 2020; <https://doi.org/10.1186/s13063-020-04828-8>). Given this reviewer comment we have sort further guidance from a meta-analysis of clinical trial SAP guidelines (Gamble et al., 2017; doi:10.1001/jama.2017.18556). We have added additional detail on this planning at the beginning of the DAP (**page 8**):

The Statistical Analysis Plan was developed based upon the UK Dept of Health/Medical Research Council Clinical Trials Toolkit and NHS epidemiological study designs, with details outlined per the standards for random control trials and clinical trials [59,60].

In response to this comment, we have also tried to build on existing text and elaborated the level of detail provided for (i) how we intend to interrogate each Hypothesis in the data analysis plan (**page 13-14**) (ii) our power calculations (**page 7**) and (iii) our overall statistical plans in the DAP (**page 8**). Although the variables of interest have been identified, statistical approaches may be adapted to include non-linear tests and sensitivity analyses where appropriate. Study design and statistical planning has been grounded in evidence from previous studies of natural disasters or major health events. These amendments to the text are evident throughout the manuscript (highlighted in grey), in this instance, given the number of changes, we have provided only one example of a substantial change to the statistical analysis plan below:

“To address Hypothesis 2, Bayesian non-linear regressions will be used to explore how COVID-19 related restrictions during an infant’s birth altered infant social exposure with caregivers and non-household social partners, and the impact of these in turn on infant processing of sounds, sights and social stimuli. The dependent variables will be derived from scores of the infant toddler sensory profile,

a standard assessment of infant self-regulation and responsiveness to their environment. On the first level, the timing of an infant's birth will be coded based on whether a lockdown or no lockdown was imposed by the government, as well as coding for more specific shifts in the UK government public health policy and 'unlocking' guidance from July 14 2020 to July 19 2021. Additional COVID-19 factors may be entered, including suspected or positive cases in loved ones, parental COVID-19 concern, pandemic-related parenting anxiety and parent-reported adherence to lockdown. On the next level, infant's social exposure will draw from caregiver-reported of their frequency of face-to-face interactions with their baby, as well as their baby's exposure to social partners from outside the household, in-person, at a distance and online. On the third level, family sociodemographic factors, such as the number of family members in each household, family income, ethnicity and high-risk health conditions will be included."

- 4. There are some recent papers discussing the intersection of parental and infant mental health, the authors may refer to the articles below to discuss some evidence and opinions in the DISCUSSION section: McVety, C.C., 2021. Effects of Parental Mental Health on Children in a Worldwide. Ye, J., 2020. Pediatric mental and behavioral health in the period of quarantine and social distancing with COVID-19. JMIR pediatrics and parenting, 3(2), p.e19867.**

Following submission guidelines provided by the journal for a 'Protocol' the student dissertation has not been included, but we appreciate the attention drawn to the JMIR paper. Along with a relevant preprint, the Ye paper has been included within the introduction on **page 1**. Furthermore, updated, more recent references have been added to manuscript, pertaining to how COVID-19 has impacted parental mental health, infant temperament and parental access to caregivers, poverty, and social care, as well as breastfeeding. Some relevant insertions summarizing the updated findings are included below:

Page 1: *"Emerging work is documenting the long-term implications of adversity related to the current pandemic including for example biological (i.e., COVID infection), acute environmental (i.e., temporary unemployment and psychosocial influences (i.e., impoverished, or atypical social environment) [14–16]. The social distancing restrictions and national lockdowns that were put in place to mitigate COVID-19 transmission have had a range of secondary consequences impacting the psychological wellbeing of pregnant women and new parents and the postnatal psychosocial environment that the infant is born into [17–19]."*

Page 3: *"For example, traumatic birth experiences amidst the changing public health situation or COVID-19 infection in the household have been associated with unusual parent-infant bonding [34]."*

Page 4: *"Lower income families reported more frequent issues with breastfeeding, higher incidence of postnatal depression, and difficulty accessing caregiving and social support that might ameliorate the demands of caring for newborns [48–50]."*

Reviewer: 2

- 1. In the introduction, the Authors refer to parental mental health, but restrict their comments solely on maternal depression and anxiety; if they talk about parental, fathers' studies should be included; in addition, mental health issues during Covid are far more complicated during the perinatal period.**

We thank the reviewer for this comment and have revised some of this passage to ensure that the language and supporting literature reflects the term parental mental health. While our existing text includes a discussion of parental anxiety, and acknowledge mood, stress, worry and health vigilance, we appreciate the reviewer's suggestion that the onset of the pandemic elicited distress akin to PTSD for some expectant and new parents. We do include a measure of how these acute effects of stress during the onset of the pandemic and/or the mother's pregnancy had an enduring impact on parental mental health, parenting and infant development. The Event Impact Scale (EIS) is included in our survey, along with scales for anxiety (STAI and PRAQ) and mood (HADS). What is lacking, we agree, is the inclusion of literature on some of these areas in our introduction. The articles that the reviewer suggests, which have been released since our submission, are now included and round out the existing literature review.

Given the substantive literature on mother's perinatal mental health, relative to fathers, we had heavily drawn on this area in our literature review. However, we agree with the reviewers that further literature on paternal mental health would be beneficial. We now expand our manuscript to include the systematic effects of the pandemic on family dynamics and caregiving responsibilities, including recent studies on first-time father's mental health and consideration of family's separations from relatives during lockdown.

Page 4: *"Similarly, first-time fathers who became parents during the onset of the pandemic in Italy reported greater stress than those with older children, and a study of fathers in Israel found that those who reported greater pandemic and parenting stress were more likely to report dysfunctional interactions with their infant and identify their baby's temperament as difficult [22,39]."*

Page 4: *"Lower income families reported more frequent issues with breastfeeding, higher incidence of postnatal depression, and difficulty accessing caregiving and social support that might ameliorate the demands of caring for newborns [48–50]."*

- 2. It would be important to make explicit the subtended theoretical model, that has guided the choice of the study variables, methods and analyses and will offer the key of data interpretation.**

We thank the reviewer for this comment. In response to this, we have included additional text on the theoretical model behind some of the choices of the study variables. Introduced in the first paragraph and now made explicit throughout the manuscript, the foetal origins of health hypothesis and developmental programming of stress response and self-regulation inform our study variables, methods and analytic approach.

Page 1: *“Infants born during periods of social disruption and disease are noted for more restricted intrauterine growth, smaller birth size, and higher lifetime incidence of chronic medical conditions such as Type-II diabetes, suggesting a role for fetal programming of endocrine dysfunction and metabolic regulation [2,3]...”*

“Conceptual frameworks have been advanced regarding the lifelong effects of adversity in pregnancy and early childhood. As specified in the developmental origins of health hypothesis, parental stress interacts with environmental exposures (e.g., nutrition, pollution), to influence the maternal-fetal physiological feedback (as indicated by hormonal and inflammatory biomarkers) [6,7].”

“Further, the ecobiodevelopmental framework illustrates how modifiable early environmental influences - such as unemployment, family poverty and access to healthcare - can impart an enduring effect on children’s stress physiology and genetic expression [6, 10, 11]. An associated framework put forward by Nelson and Gabard-Durnam [12,13] suggests that we should view adversity as a violation of the expectable environment, with emphasis placed on the magnitude of this impact being greater during critical periods of brain development (such as the first 1000 days from conception to toddlerhood).”

Further to this, we have linked the conceptual framework to the hypotheses in the Study Design on **page 5-6:**

“Our variable selection and sequential building of hypotheses embed our key frameworks: i) examining parental mental health in light of stress and social, financial and contextual factors (ecobiodevelopmental model); ii) how infant sensory processing pertains to caregiving and social exposure of infants, relative to lockdown/COVID-19 transmission during infant’s birth, family COVID-19 vigilance and parenting anxiety (early expectable environment); iii) the interaction of maternal mental health and fetal growth measures as longitudinal predictors of infant cognitive outcomes (developmental origins of disease hypothesis); iv) finally, encompassing the social, financial and contextual factors which impact parental mental health to shape infant temperament and sensory processing, accounting for early infant caregiving and social environment (developmental programming). Ultimately, our research programme can demonstrate support (or lack thereof) for extending developmental programming-based frameworks beyond child physical health (insulin resistance, stature, etc.) [54,55] and cognitive outcomes [56–58], to explain variability in proximal domains such as infant affective, social and sensory capacities.”

VERSION 2 – REVIEW

REVIEWER	Vismara, Laura University of Cagliari
REVIEW RETURNED	20-Jan-2022
GENERAL COMMENTS	Thank you for your careful revisions. I now endorse the work for publication.